# Smoking and pre-existing co-morbidities as risk factors for developing severity of COVID-19 infection: Evidence from a field hospital in a rural area of Bangladesh

Rashadul Islam[1]*, Sayem Ahmed[2], Samar Kishor Chakma[1]*, Tareq Mahmud[1], Abdullah Al Mamun[1], Ziaul Islam[1], M. Munirul Islam[1]

1 International Centre for Diarrhoeal Disease Research, Bangladesh (icddr,b), Dhaka, Bangladesh, 2 Health Economics and Health Technology Assessment, Institute of Health and Wellbeing, University of Glasgow, Glasgow, United Kingdom

* md.rashadul@icddrb.org (RI); samar.chakma@icddrb.org (SKC)

**Data Availability Statement:** All relevant data are within the paper and its Supporting information files.

## Abstract

Since August 2020; the International Centre for Diarrhoeal Disease Research, Bangladesh (icddr,b) in collaboration with UNICEF has been operating a COVID-19 field hospital at the Teknaf sub-district of Cox's Bazar in Bangladesh. This paper is focused on estimating the effects of a history of tobacco smoking and pre-existing co-morbidities on the severity of COVID-19 infection among adult patients admitted into the aforesaid hospital. We conducted a retrospective data analysis of COVID-19 adult patients hospitalized from August 27, 2020 to April 20, 2022. Based on inclusion criteria; a total of 788 admitted patients were included in the analysis. We conducted a Chi-squared test and Fisher's exact test for the categorical variables to see their associations. Multinomial logistic regression models were performed to explore the risk factors for the severity of COVID-19 infection. Among 788 patients, 18.4%, 18.8%, 13%, 7.1%, 3.4%, and 1.9% have had a history of smoking, hypertension, diabetes, chronic obstructive pulmonary disease (COPD), cardiovascular diseases (CVD), and asthma respectively. Overall, the mean age of the patients was 40.3 ± 16.4 years and 51% were female. In multivariate analysis, history of smoking and co-morbidities were identified as the risk factors for the severity of COVID-19 infection; the history of smoking was found linked with an increase in the risk of developing critical, severe, and moderate level of COVID-19 infection- notably 3.17 times (RRR = 3.17; 95% CI: 1.3–7.68), 2.98 times (RRR = 2.98; 95% CI: 1.87–4.76) and 1.96 times (RRR = 1.96; 95% CI: 1.25–3.08) respectively more than the patients who never smoked. It was evident that patients with at least one of the selected co-morbidities such as hypertension, diabetes, COPD, CVD, and asthma exhibited a significantly higher likelihood of experiencing severe illness of COVID-19 compared to patients without any co-morbidity. History of tobacco smoking and pre-existing co-morbidities were significantly associated with an increased severity of COVID-19 infection.

**Funding:** The author(s) received no specific funding for this work.

# Background

## The pandemic

COVID-19 (Coronavirus Disease, 2019) was first detected in Wuhan province of China in early December 2019 [1]. It is a highly infectious, rapidly transmissible and fatal disease caused by Severe Acute Respiratory Syndrome Coronavirus 2 (SARS-CoV-2) [2]. Globally, it infected over 587 million people, including 6.44 million deaths till August 10, 2022 [3]. COVID-19 outbreak was declared a Public Health Emergency of Global Concern on January 30, 2020, and a pandemic on March 11, 2020, by the World Health Organization (WHO) [1]. In Bangladesh, the first 3 COVID-19 cases were detected in March 2020 [4] and more than 2.0 million COVID-19 cases were identified, including about 29,300 deaths till August 10, 2022 [5].

## Literature review

Globally, several studies have been conducted to identify the factors associated with COVID-19 disease severity, with most of the studies focusing on a limited number of conditions (i.e., comorbidities) including smoking status [5–8]. Approximately 20% of adults worldwide smoke tobacco, which is recognized as a significant risk factor for respiratory diseases [6, 9] and smokers have a 34% greater risk of contracting the common flu than among non-smokers [10]. Also, smoking leads to inflammation of the pulmonary epithelium causing cytokines release, increased mucous secretion, and impaired mucociliary clearance [11, 12]. The prevalence of tobacco use in Bangladesh is significant, with nearly 35% of adults primarily smoking tobacco currently [13]. Since 2004, this lower middle-income country has witnessed a dramatic increase in the health and economic burden linked to smoking or smokeless tobacco use, as evidenced by primary data from a nationally representative survey of patients with tobacco-related illnesses, where tobacco smoking significantly contributes to non-communicable diseases (NCDs), including cardiovascular and respiratory conditions, underscoring the imperative to address tobacco control measures to mitigate its impact on both public health and economic growth [14].

Several studies have found mixed results in regard to the association between smoking and the severity of COVID-19 disease [5, 7]. An analysis of a systematic review and meta-analysis indicated that current and former smokers are more likely to present with severe COVID-19 conditions as compared with non-smokers [15], while another study reported that the severity of COVID-19 illness was not associated with smoking [7]. Additionally, smoking tobacco is strongly linked to the development of co-morbidities such as hypertension, cardiovascular disease (CVD), chronic obstructive pulmonary disease (COPD), and asthma, which may worsen the outcome of COVID-19 disease [7, 17].

Hypertension is one of the most common and frequent co-morbidities which may contribute to the development of acute respiratory distress syndrome in patients with COVID-19 positivity [16]. Various findings have been reported regarding the association between the comorbidity of hypertension and the severity of COVID-19 [17–19]. According to a retrospective cohort study, hypertension was not an independent risk factor but was linked to diabetes and cardiovascular disease [20].

Diabetes, another common comorbidity, has been identified as a significant public health issue in Bangladesh and affects approximately one in ten adults [21]. A study found that diabetic patients with COVID-19 had a more severe condition and poorer clinical outcomes, as well as a higher mortality rate and increased ICU admissions, highlighting the need for improved management strategies in order to improve their prognosis [22–24]. Nevertheless, a

study conducted in China with a limited sample size did not find diabetes to be associated with severe COVID-19 [25].

Globally, approximately 200 million people suffer from COPD, and 3.2 million people die from COPD every year, making it the third leading cause of death among adults [26], and currently, 12.5% of Bangladeshi adults have COPD [27]. People with chronic obstructive pulmonary disease can suffer from acute deterioration of their health status due to respiratory viruses, and since the start of the COVID-19 pandemic, they have faced additional challenges [28]. Several studies have demonstrated that patients with underlying COPD are more likely to experience severe COVID-19 outcomes (e.g., hospitalization, severe disease, and ICU admission) [6, 29]. However, a study conducted on symptomatic patients found that COPD and asthma alone did not lead to worsening infections, hospitalizations, or deaths associated with COVID-19 [30]. Similarly, a study found cases with CVD and asthma show a significantly elevated risk of increasing severity of COVID-19 [31].

Despite several studies in many countries, to date, only one study has been found in Bangladesh which focused on the association between the smoking habit and the severity of COVID-19 infection but did not consider any specific co-morbidities [5]. To address this evidence gap; our objective was to estimate the effects of a history of smoking, either cigarettes or other tobacco products and pre-existing co-morbidities on the severity of COVID-19 infection in a rural setting of Bangladesh.

## Materials and methods

### The study setting

The hospital, initiated as part of COVID-19 outbreak management in Teknaf, began operations on August 27, 2020, and it is still actively providing services to the local population and Forcibly Displaced Myanmar Nationals (FDMN) as a not-for-profit organization. Out of total 95 beds; initially, 80 beds of this field hospital were dedicated to COVID-19 patients inclusive of a 10-bed High Dependency Unit (HDU) with central oxygen, concentrator, and cylinder oxygen supply. Besides; 24/7 Emergency, Out-Patient Department, radiology, laboratory services, and referral by ambulance to higher level facilities in Cox's Bazar district remain available free of charge. Necessary medicines, high flow nasal cannula, pulse oximeter, PPE, masks, hand sanitizer, and other necessary medical appliances were adequately supplied by the donor. A large contingent of trained medical doctors, nurses, patient care attendants, medical technologists, infection prevention and control (IPC) workers, and biomedical engineers were deployed by the International Centre for Diarrhoeal Disease Research, Bangladesh (icddr,b) to run the facility effectively. An uninterrupted supply of power and safe water was ensured in the hospital. However; the physical structure of the hospital was made of bamboo huts with concrete floors.

### Study design and data source

We conducted a retrospective analysis of the data collected from COVID-19 positive adult patients who were hospitalized at Teknaf field hospital. Upon initial contact, the physician used a standardized Case Record Form (CRF) to record the patient's background characteristics, medical history, smoking history (current or past smoker), co-morbidity, the severity of illness and underlying co-morbidities. A skilled data management team maintains and updates the hospital's online database system named "ONA" to accommodate the collection of patient information. We extracted data from COVID-19 patients treated between August 27, 2020 and April 20, 2022 (in .CSV format). There were 961 patients admitted to the study hospital

during this period who were diagnosed as positive for COVID-19 by both nasal and oropharyngeal swabs in Real-Time Polymerase Chain Reaction (RT-PCR).

Because this study focused on adult patients (aged 18 years or older or older), 173 patients under the age of 18 were excluded from this analysis. We chose to categorize patients as "adults" (18–60 years) and "elderly" (>60 years) [32] due to the observed significant association between these age groups and the severity of COVID-19 infection in the pilot analysis. Therefore, finally, 788 COVID-19 patients were included in this analysis. COPD, hypertension, cardiovascular disease, and diabetes were diagnosed based on patients' history, clinical features and radiological and laboratory findings. For this study, the current and past smokers were grouped as 'ever-smoker'.

## The severity of COVID-19 infection

The severity of COVID-19 infection was determined based on WHO standards [33]. The severity levels of COVID-19 infection (only for adult patients) are explained as,

- Mild type: Clinical symptoms (e.g., fever, cough, fatigue, anorexia, shore throat, nasal congestion, headache, diarrhea, nausea and vomiting, dizziness, weakness) are present, without evidence of viral pneumonia or hypoxia (Hypoxia: New-onset oxygen saturation of $\leq 94.0\%$).

- Moderate type: clinical signs of pneumonia (fever, cough, dyspnea, fast breathing) and other non-specific symptoms (sore throat, nasal congestion, headache, diarrhea, nausea, and vomiting) are present but no signs of severe pneumonia, including $SpO_2 \geq 90\%$ on room air.

- Severe type: Patients with at least one of the following: $SpO_2 < 90\%$ on room air; Severe pneumonia; signs of severe respiratory distress (in adults, accessory muscle use, inability to complete full sentences, respiratory rate > 30 breaths/min;);

- Critical type: Acute Respiratory Distress Syndrome (ARDS), sepsis, septic shock is present, and the need for invasive or non-invasive mechanical ventilation or vasopressor therapy.

During the course of treatment, the duty physician assigned everyone included in this study a severity level based on the above definition. It was used in the analysis as a dependent variable.

It is important to note that in this study, both cigarette smoking and other tobacco product use were considered as part of smoking, encompassing a broad spectrum of tobacco consumption habits.

## Statistical analysis

The data was analyzed using STATA 16.0 (StataCorp LLC) statistical package and descriptive statistics were used to determine the frequency and percentage of background, characteristics, severity levels of COVID-19 infection, pre-existing comorbidities, and smoking history. To assess the association between main variables of interest (e.g., history of smoking, underlying co-morbidities) and severity of COVID-19 infection, Pearson's Chi-square test and Fisher's exact test are used. Multinomial logistic regression models [34] were used to identify potential risk factors and determine the effect size of these factors on the progression of COVID-19 infection severity in the patients. To measure statistical significance, a P-value of 0.05 was considered.

The MNL model is described below:

There are k categorical outcomes (e.g., mild, moderate, severe, and critical) and—without loss of generality—let the base outcome be 1. The probability that the response for the $j^{\text{th}}$

observation is equal to the $i^{th}$ outcome is

$$p_{ij} = Pr(y_j = i) = 1/(1 + \sum\nolimits_{m=2}^{k} exp(X_j\beta_m)), \text{ if } i = 1 \tag{1}$$

$$p_{ij} = Pr(y_j = i) = exp(X_j\beta_m)/(1 + \sum\nolimits_{m=2}^{k} exp(X_j\beta_m)), \text{ if } i > 1 \tag{2}$$

With k = 1, 2, 3, 4

where $x_j$ is the row vector of observed values of the independent variables for the $j^{th}$ observation and $\beta_m$ is the coefficient vector for outcome $m$.

Eq (1) used to measure the associations of severity of COVID-19 infection with the variables of interest in separate unadjusted models. The following Eq (2) was estimated to examine the relationship between the severity of COVID-19 infection and the history of smoking with control variables such as age, sex, hypertension, diabetes, COPD, CVD, and asthma. To assess the goodness-of-fit of the model, we used the likelihood ratio chi-square test, and the McFadden's pseudo-R-squared.

### Ethical clearance

The Ethical Review Committee (ERC) of the International Center for Diarrheal Disease Research, Bangladesh, waived the requirement for Institutional Review Board (IRB) review since all data is de-identified and no personal information will be disclosed. The participants gave verbal consent for their information to be used for further research since they did not intend to sign any documents or leave fingerprints, and many of them were analphabetic. Furthermore, no minors were included in this study, hence parent or guardian consent was not required.

## Results

### Patient characteristics

A total of 788 COVID-19 positive hospitalized patients were studied, of which 3.4% were critical, 20.6% were severe, 29.4% were moderate and 46.6% were mild. Overall, the mean age of the patients was 40.3 ± 16.4 years and 13.0% of patients were aged above 60 years. The majority of patients were female, with 51.0% being female, and 18.4% being ever-smokers. Based on the selected co-morbidities, 18.8% of patients had hypertension, 13.0% had diabetes, 7.1% had COPD, 3.4% had cardiovascular disease, and 1.9% had asthma as shown in Table 1.

### Severity of COVID-19 infection and patient's characteristics

Table 2 presents bivariate associations between the severity of the COVID-19 infection and the characteristics of the patients. There was a significant association (P<0.05) between the age and smoking history of the patients and COVID-19 infection. The percentage of severely COVID-19 infection was higher among the elderly patients (60 and older), compared to relatively younger adults. The proportions of patient with moderate, severe, and critical COVID-19 infection were higher among ever-smoker in comparison to non-smoker.

### Severity of COVID-19 infection and pre-existing co-morbidities

Table 3 presents the outcomes of a bivariate analysis of the severity of COVID-19 infection and selected pre-existing co-morbidities (e.g., hypertension, diabetes, COPD, CVD, asthma). There was a significant association between the presence of these co-morbidities and the

**Table 1. Distribution of patient characteristics and co-morbidities in COVID-19 cases.**

| Characteristics | Category | n (%) |
|---|---|---|
| **Sex** | Female | 402 (51.0) |
| | Male | 386 (49.0) |
| **Age (years), mean ± SD [95%Cl]** | 40.3 ± 16.4 [23.9–56.7] | |
| **Age** | 18–60 years (adult) | 686 (87.0) |
| | >60 years (elderly) | 102 (13.0) |
| **History of Smoking** | Ever-smoker | 145 (18.4) |
| | Non-smoker | 643 (81.6) |
| **Severity of COVID-19 infection** | Mild | 367 (46.6) |
| | Moderate | 232 (29.4) |
| | Severe | 162 (20.6) |
| | Critical | 27 (3.4) |
| **Co-morbidities** | Yes | 241 (30.6) |
| **Hypertension** | Yes | 148 (18.8) |
| | No | 640 (81.2) |
| **Diabetes** | Yes | 103 (13.0) |
| | No | 685 (87.0) |
| **Chronic obstructive pulmonary disease (COPD)** | Yes | 56 (7.1) |
| | No | 732 (92.9) |
| **Cardiovascular disease (CVD)** | Yes | 27 (3.4) |
| | No | 761 (96.6) |
| **Asthma** | Yes | 15 (1.9) |
| | No | 773 (98.1) |
| **Total** | | **788 (100)** |

severity of COVID-19 infection. 50.0% of patients with COPD developed severe COVID-19 infection, followed by moderate (30.4%), and critical (16.0%) cases. Among hypertensive patients, the highest proportion (50.7%) had moderate infection, then severe (37.2%), and critical (6.1%). Patients with diabetes, CVDs, and asthma had similar patterns of severity of COVID-19 infection (i.e., the highest proportion had a moderate level of severity followed by severe and critical conditions).

**Table 2. Distribution and association of patient characteristics with severity of COVID-19 infection.**

| Characteristics | Mild n (%) | Moderate n (%) | Severe n (%) | Critical n (%) | P value |
|---|---|---|---|---|---|
| **Sex** | | | | | |
| Male | 193 (50.0) | 107 (27.7) | 76 (19.7) | 10 (2.6) | 0.213[b] |
| Female | 174 (43.3) | 125 (31.1) | 86 (21.4) | 17 (4.2) | |
| **Age** | | | | | |
| 18–60 years (adult) | 360 (52.5) | 195 (28.4) | 118 (17.2) | 13 (1.9) | 0.001[a] |
| >60 years (elderly) | 7 (6.8) | 37 (36.3) | 44 (43.1) | 14 (13.7) | |
| **History of Smoking** | | | | | |
| Ever-smoker | 43 (29.7) | 48 (33.1) | 46 (31.7) | 8 (5.5) | 0.001[b] |
| Non-smoker | 324 (50.4) | 184 (28.6) | 116 (18.0) | 19 (3.0) | |

[a] Fisher's exact test; p-value = .0000 treated as 0.001

[b] Chi-square test; p-value = .0000 treated as 0.001

**Table 3. Association between the co-morbidities and severity of COVID-19 infection.**

| Co-morbidities | Status | Mild n (%) | Moderate n (%) | Severe n (%) | Critical n (%) | P-value |
|---|---|---|---|---|---|---|
| **Hypertension** | Yes | 9 (6.0) | 75 (50.7) | 55 (37.2) | 9 (6.1) | 0.001[b] |
| | No | 358 (55.9) | 157 (24.5) | 107 (16.7) | 18 (2.8) | |
| **Diabetes** | Yes | 4 (3.8) | 51 (49.5) | 38 (37.0) | 10 (9.7) | 0.001[a] |
| | No | 363 (53.0) | 181 (26.4) | 124 (18.1) | 17 (2.5) | |
| **Chronic obstructive pulmonary disease (COPD)** | Yes | 2 (3.6) | 17 (30.4) | 28 (50.0) | 9 (16.0) | 0.001[a] |
| | No | 365 (49.8) | 215 (29.4) | 134 (18.3) | 18 (2.5) | |
| **Cardiovascular Disease (CVD)** | Yes | 2 (7.4) | 13 (48.2) | 9 (33.3) | 3 (11.1) | 0.001[a] |
| | No | 365 (47.9) | 219 (28.7) | 155 (20.1) | 24 (3.1) | |
| **Asthma** | Yes | 2 (13.3) | 8 (53.4) | 5 (33.3) | No observation | 0.031[a] |
| | No | 365 (47.2) | 224 (29.0) | 157 (20.3) | 27 (3.5) | |

[a] Fisher's exact test; p-value = .0000 treated as 0.001

[b] Chi-square test; p-value = .0000 treated as 0.001

## Regression results

Table 4 presents the results of the multinomial logistic regression analysis of the effects of potential risk factors on the progression of severity of COVID-19 infection among the admitted patients. The unadjusted model indicated that ever-smoker COVID-19 patients suffered 3.17 (RRR = 3.17; 95% CI: 1.3–7.68) times more critical stages of the COVID-19 disease severity in comparison with non-smoker COVID-19 patients. Compared to non-smoker, the same group of patients suffered 1.96 (RRR = 1.96; 95% CI: 1.25–3.08) times more moderate and 2.98 (RRR = 2.98; 95% CI:1.87–4.76) times more severe COVID-19 infections. Similarly, co-morbidities were identified as risk factors for critical, severe, and moderate severity levels of COVID-19 infection in the separate unadjusted models.

As potential risk factors, the following variables were incorporated into the final model (adjusted model): smoking history, hypertension, diabetes, COPD, CVD, asthma, gender, and age (>60 years old—elderly). In the adjusted model, while smoking history was recognized as a risk factor contributing to the severity of COVID-19 infection, it was not associated with moderate and critical levels of severity. Hypertensive patients were 4.75 times more proneto experience critical severity of COVID-19 infection (RRR = 4.76; 95% CI: 1.47–15.04) compared to non-hypertensive patients. In addition, hypertensive patients were 8.51 times more vulnerable to develop severe infections (RR = 8.51; 95% CI: 3.80–19.08) and 9.13 times more vulnerable to suffering moderate infections (RR = 9.13; 95% CI: 4.25–19.59) compared to those without hypertension. Similarly, patients with diabetes had 42.29 times higher risk of developing critical COVID-19 infections (RRR = 42.29; 95% CI: 10.65–167.93), 16.2 times higher risk of severe infections (RRR = 16.2; 95% CI: 5.39–48.96), and 13.19 times higher risk of moderate infections (RRR = 13.19; 95% CI: 4.5–38.69) compared to patients without diabetes. Patients with COPD were 36.55 times susceptible to experience critical COVID-19 infection (RRR = 36.55; 95% CI: 6.45–207.1), 18.78 times susceptible to have severe infections (RRR = 18.78; 95% CI: 4.12–85.48), and 8.45 times susceptible to have moderate infections (RRR = 8.45; 95% CI: 1.82–39.13) when compared with non-COPD patients. Considering asthma, patients with this condition were over 8 times risk of having worse COVID-19 infection conditions of moderate to severe. As a result of the limited number of asthmatic participants in the study (n = 15), no association was established between pre-existing asthma and the development of critical stages of COVID-19 infection. Lastly, elderly patients (>60 years) had 42.83 times critical COVID-19 infection (RRR = 42.83; 95% CI: 13.2–138.9), had12.35 times severe infections (RRR = 12.35;

**Table 4. Factors associated with severity of COVID-19 infection using multinomial logistic regression models.**

| Outcome variable | Disease severity | Ref. category | Unadjusted Models RRR (95% CI) | Adjusted Model RRR (95% CI) |
|---|---|---|---|---|
| | Mild | (Base outcome) | | |
| **History of Smoking** | Moderate | Ever-smoker (Ref: Never-smoker) | 1.96 (1.25–3.08)*** | 1.55 (0.91–2.64) |
| | Severe | Ever-smoker (Ref: Never-smoker) | 2.98 (1.87–4.76)*** | 1.98 (1.11–3.56)** |
| | Critical | Ever-smoker (Ref: Never-smoker) | 3.17 (1.30–7.68)** | 1.52 (0.50–4.54) |
| **Hypertension** | Moderate | Hypertension (Ref: No hypertension) | 19 (9.28–38.89)*** | 9.13 (4.25–19.59)*** |
| | Severe | Hypertension (Ref: No hypertension) | 20.44 (9.78–42.73)*** | 8.51 (3.80–19.08)*** |
| | Critical | Hypertension (Ref: No hypertension) | 19.88 (7.04–56.16)*** | 4.76 (1.47–15.04)*** |
| **Diabetes** | Moderate | Diabetes (Ref: No diabetes) | 25.57 (9.09–71.85)*** | 13.19 (4.5–38.69)*** |
| | Severe | Diabetes (Ref: No diabetes) | 27.81 (9.72–79.49)*** | 16.2 (5.39–48.96)*** |
| | Critical | Diabetes (Ref: No diabetes) | 53.3 (15.18–187.7)*** | 42.29 (10.65–167.93)*** |
| **Chronic Obstructive Pulmonary Disease (COPD)** | Moderate | COPD (Ref: No COPD) | 14.43 (3.30–63.06)*** | 8.45 (1.82–39.13)*** |
| | Severe | COPD (Ref: No COPD) | 38.1 (8.96–162.26)*** | 18.78 (4.12–85.48)*** |
| | Critical | COPD (Ref: No COPD) | 91.2 (18.35–453.5)*** | 36.55 (6.45–207.1)*** |
| **Cardiovascular Disease (CVD)** | Moderate | CVD (Ref: No CVD) | 10.83 (2.42–48.45)*** | 4.98 (0.93–26.47)* |
| | Severe | CVD (Ref: No CVD) | 10.73 (2.29–48.45)*** | 3.93 (0.67–23.01) |
| | Critical | CVD (Ref: No CVD) | 22.81 (3.63–143.1)*** | 5.29 (0.56–49.3) |
| **Asthma** | Moderate | Asthma (Ref: No Asthma) | 6.5 (1.37–30.91)** | 8.15 (1.63–40.65)** |
| | Severe | Asthma (Ref: No Asthma) | 5.8 (1.11–30.23)** | 8.52 (1.54–47.20)** |
| | Critical | Asthma (Ref: No Asthma) | No observation | No observation |
| **Sex** | Moderate | Female (Ref: Male) | 1.29 (0.93–1.8) | 1.10 (0.74–1.62) |
| | Severe | Female (Ref: Male) | 1.25 (0.86–1.81) | 1.24 (0.78–1.95) |
| | Critical | Female (Ref: Male) | 1.88 (0.84–4.22) | 2.27 (0.88–5.81) |
| **Age** | | | | |
| **>60 yr.(elderly)** | Moderate | >60 yr. (Ref: 18–60 yr.) | 9.75 (4.27–22.29)*** | 7.33 (3.07–17.48)*** |
| | Severe | >60 yr. (Ref: 18–60 yr.) | 19.17 (8.41–43.72)*** | 12.35 (5.13–29.73)*** |
| | Critical | >60 yr. (Ref: 18–60 yr.) | 55.38 (19.13–160.29)*** | 42.83 (13.2–138.9)*** |
| **Number of obs.** | | | | 788 |
| **LR chi2(6)** | | | | 338.2 |
| **Prob > chi2** | | | | 0.001 |
| **Pseudo R2** | | | | 0.1855 |
| **Log likelihood** | | | | -742.37 |

*p<0.1;

**p<0.05;

***p<0.01;

p-value = .0000 treated as 0.001

95% CI: 5.13–29.73), and had 7.33 times moderate infections (RRR = 7.33; 95% CI: 3.07–17.48) than mild, when compared to adults aged between 18 and 60 years.

## Discussion

The present study investigated the vulnerability to severe COVID-19 infection in smokers of both sexes, especially those with comorbidities like hypertension, diabetes, COPD, CVD, and

asthma. Age was also identified as a significant factor in determining the severity of COVID-19 infection. Multiple factors were investigated that contributed to the severity of COVID-19 infection independently and cumulatively. The prevalence of ever-smoker among the studied patients was approximately eighteen percent, with both females and males being ever-smoker, though the prevalence was not examined separately. Additionally, a subset of patients (30.6%) presented with at least one of the selected underlying co-morbidities.

The study revealed a significant association between a history of smoking and the severity of COVID-19 infection where patients with smoking habits were at a greater risk of developing severe, and critical stages of COVID-19 infection than patients who did not smoke. This finding was supported by a meta-analysis which found that current or former smokers had a significantly higher chance of developing severe or critical cases of COVID-19 infection. Another study conducted in Dhaka, Bangladesh showed that people who had actively smoked for a long time were more likely to develop severe and critical COVID-19 [5]. Additionally, there is evidence that smokers have a greater risk of poor outcomes from COVID-19, such as hospitalization and progression to severe disease [35]. The effects of smoking may be attributed to the significant changes in cytokines that can affect systemic immunity and inflammation [36] since cytokines play a crucial role in the pathogenesis of COVID-19 [37]. Furthermore, smokers are vulnerable to get common cold and respiratory viral diseases [11]. Lippi G and Henry BM, however, found that smoking did not contribute to the severity of COVID-19 illness in their study [7].

Although a retrospective cohort study found that, hypertension was not an independent risk factor for COVID-19 severity [20]. However, current study found that hypertension is significantly associated with the severity of COVID-19 infection. Even after adjusting for other factors, it was identified as a potential risk factor attributable to moderate, severe, and critical stages of COVID-19 infection. This finding is consistent with another systematic review and meta-analysis findings which revealed that hypertension was associated with an increased composite poor outcome (risk ratio (RR) 2.11) and its subgroups, including mortality, severe COVID-19, ARDS, intensive care, and disease progression [38]. Moreover, another systematic review identified hypertension as the most prevalent comorbidity in COVID-19 patients which noticeably increases the risk of hospitalization and death [39].

Likewise, diabetes and asthma were also found to be risk factors for COVID-19 infection severity in the current study, with the adjusted odds of severe infection being higher among those with diabetes and asthma. There was, however, no significant association between diabetes and severe COVID-19 outcomes in a study conducted in China with a limited sample size [25]. This study's findings, however, may not be easily generalizable to wider populations or contexts due to the limitations of its sample size. The current study also found diabetes, and asthma as risk factors for the severity of COVID-19 infection. Similarly, a study conducted in the United States of America found that diabetes and asthma are associated with an increased severity of COVID-19 infection [31]. Additionally, asthma has been identified as a potential comorbid condition associated with severe illness of COVID-19 [34] by the Centers for Disease Control and Prevention [40].

Although univariate analysis found that two-thirds of the patients with cardiovascular diseases were severely ill and 11% were critically ill, the multivariate analysis adjusted with covariates found CVD to be insignificant as an independent predictor of COVID-19 infection severity. Kunihiro et al similarly, found that CVD was independently associated with COVID-19 severity [19].

The presence of underlying COPD among 75 percent of patient was significantly associated with a greater risk of developing severe infection with COVID-19. Multivariate analysis after adjusting for the covariates such as age, gender, smoking history, hypertension, diabetes,

CVD, and asthma revealed that this association persisted, indicating COVID-19 patients with COPD had worse conditions than those without COPD. As highlighted in a narrative review of the basic science and clinical outcomes of COVID-19, COPD is associated with an increased risk of severe illness independently [41]. Moreover, a study examined the distinct effects of asthma and COPD on the progression and outcomes of COVID-19 patients and found that patients suffering from COPD were 23 times more likely to develop severe illness than patients without COPD [42]. Nevertheless, a study conducted on symptomatic patients found that COPD and asthma alone did not worsen infections, hospitalizations, or deaths associated with COVID-19 [30].

Patients' age was identified as a significant risk factor associated with hospitalization for COVID-19 critical illness. Elderly patients, particularly those aged 60 years and older, had an increased risk of severe COVID-19-associated outcomes. This finding aligned with the data reported by the Centers for Disease Control and Prevention which found that elderly patients with COVID-19 infection were more likely to suffer severe COVID-19-associated outcomes [40]. Nevertheless, future large-scale research is warranted to address the lack of conclusive estimation of severe rates, thereby enhancing our understanding of COVID-19 severity and elderly age [43].

The prevalence of undiagnosed hypertension, diabetes, and COPD is high-burden in Bangladesh [44–46], and a majority of the population is unaware of these diseases and their consequences. Based on the Bangladesh Demographic and Health Survey (BDHS), 50% of hypertensive women and 66% of hypertensive men are unaware of their high blood pressure, and 60% of diabetic patients are unaware of their elevated blood glucose levels among adults (aged 18 and older) [47]. During the course of this study, we gained insight into the local people's lifestyle, attitudes, and responses to diseases, as well as the environmental factors that contribute to disease. In Teknaf, the prevalence of smoking and undiagnosed non-communicable diseases (NCDs) is particularly high, especially among respiratory patients. While the patients in our study were tested for these co-morbidities during their hospital stay, there is a possibility that some patients may not have been aware of their pre-existing conditions. This particularly applies to first-time patients, like the Forcibly Displaced Myanmar Nationals (FDMN) refugees, who might lack prior medical records.

## Strength of the study

The main strength of this study is the use of existing dataset which covers all the three waves of COVID-19 surges in a remote rural setting of Bangladesh. All the stages of severity of COVID-19 infection were well diagnosed by the trained physicians according to the pre-established standard definitions set by the National Guidelines on COVID-19 Management. Therefore, the dataset was well representative of the defined severity of COVID-19 infection in Bangladesh.

## Study limitations

The vaccination uptake and its potential impact on moderating the severity of COVID-19 infection were not considered in the analysis. Also, the clinical team did not perform spirometry to diagnose COPD. However, physicians diagnosed the patients by clinical examination and X-ray chest in the hospital. Study patients were unwilling to share their monthly income and occupation for which this study could not capture their socio-economic status. Furthermore, while we examined the individual effects of sex and smoking on the severity of COVID-19 infection, we did not conduct a specific analysis to explore potential interaction effects between these factors that could influence the development of severe COVID-19.

## Conclusions

It is evident from the study findings that smoking history and underlying co-morbidities are risk factors for the development of COVID-19 infection severity. Among the covariates of COVID-19 infection severity, the age of patients also contributes to worsening the severity status. Policymakers, public health activists, and clinical practitioners need to take advantage of these findings to plan interventions that can contribute to improving the preventive strategy for reducing the above-mentioned risk factors and the management of those risk factors in clinical settings. Also, it is recommended that public health initiatives prioritize extensive smoking cessation programs and awareness campaigns aimed at reducing smoking rates, particularly among individuals with pre-existing health conditions. Additionally, enhancing diagnostic infrastructure for those with comorbidities can aid in the timely identification of their health issues.. Furthermore, efforts should be made to enhance the early detection and management of hypertension, diabetes, COPD, and asthma in the community to mitigate the severity of COVID-19 infections and improve overall health outcomes.

## Supporting information

**S1 Data.**
(CSV)

## Acknowledgments

We would like to express our sincere thanks to the Data Management team, Laboratory Technologists, and Radiology Technologists of SARI ITC Teknaf for providing their continuous support in cleaning and compiling data. Heartful thanks to the Management staffs, Doctors specially Dr. Sojol Jahangir Alom, and Nurses of SARI ITC for their cordial co-operation and support. icddr,b is also grateful to the Governments of Bangladesh, Canada, Sweden and the UK for providing core/unrestricted support.

## Author Contributions

**Conceptualization:** Rashadul Islam, Ziaul Islam.

**Data curation:** Rashadul Islam, Tareq Mahmud, Abdullah Al Mamun.

**Formal analysis:** Rashadul Islam, Sayem Ahmed.

**Methodology:** Rashadul Islam, Sayem Ahmed.

**Resources:** M. Munirul Islam.

**Supervision:** Sayem Ahmed, Samar Kishor Chakma, Ziaul Islam, M. Munirul Islam.

**Validation:** M. Munirul Islam.

**Visualization:** Rashadul Islam.

**Writing – original draft:** Rashadul Islam.

**Writing – review & editing:** Sayem Ahmed, Samar Kishor Chakma, Ziaul Islam, M. Munirul Islam.

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
