## [Editor Report · Decision Letter 0]

20 Mar 2023

PONE-D-22-33469Smoking and Pre-existing Co-morbidities as Risk Factors for Developing Severity of COVID-19 Infection: Evidence from a Field Hospital in a Rural Area of BangladeshPLOS ONE

Dear Dr. Islam,

Thank you for submitting your manuscript to PLOS ONE. After careful consideration, we feel that it has merit but does not fully meet PLOS ONE’s publication criteria as it currently stands. Therefore, we invite you to submit a revised version of the manuscript that addresses the points raised during the review process.

We look forward to receiving your revised manuscript.

Kind regards,

Md. Feroz Kabir, BPT, MPT, MPH, BPED, MPED

Academic Editor

PLOS ONE

Journal Requirements:

Additional Editor Comments :

Please clearly mentioned the sampling procedures and the population and calculation of sample in Methodology section.

Give very early and current study synchronization in Literature review section.

Clearly state the disease severity and patient’s characteristics and properly acknowledge it in Result section

Compare and contrast with the current available evidence.

Thanks for your hard work.

---

## [Author Response · Author response to Decision Letter 0]

17 Apr 2023

Sampling procedures and the population: We explained clearly the sampling procedures and the study population in the “Study design and data source” in Materials and methods section.

Sample size calculation: According to the inclusion and exclusion criteria, this study included all the patients who were COVID-19 positive and admitted at the studied hospital. This is a retrospective analysis of medical records of the patients. Therefore, the calculation of sample size could be considered as a waive. 

For example; the below listed studies also did not consider sample size calculation in their retrospective methodology. 

1. Adrish M, Chilimuri S, Mantri N, et al. Association of smoking status with outcomes in hospitalised patients with COVID-19. BMJ Open Resp Res 2020;7:e000716. doi:10.1136/ bmjresp-2020-000716; https://bmjopenrespres.bmj.com/content/bmjresp/7/1/e000716.full.pdf

 2. Asthma and COPD Are Not Risk Factors for ICU Stay and Death in Case of SARS-CoV2 Infection; https://doi.org/10.1016/j.jaip.2020.09.044

 3. Clinical Characteristics of Coronavirus Disease 2019 in China, N Engl J Med 2020;382:1708-20.DOI: 10.1056/NEJMoa2002032; https://www.nejm.org/doi/10.1056/nejmoa2002032

Other changes:

I have now added the following keywords: Severity of COVID-19; SARS-CoV2; Risk factors; Smoking history; Co-morbidities; icddr,b; Bangladesh;” just after the abstract section (line-46).

I have rephrased the “Introduction” to “Background” (line-48), “COVID-19 disease severity” and “severity of COVID-19 disease” to “severity of COVID-19 infection” throughout the manuscript. 

Literature review section is now up-to-date with the clear and current study synchronization. Below are the changes in reference list:

In the pandemic (line 49): Ref. #4 is now moved to the ref. # 1 and ref. #3 move to the ref. #5; ref. #2, #3, and #4 is newly inserted; 

In Literature review (line 58): Ref. #5, #1, #11, #12, #7, #30, #16, #17, #18, #19, #21, #22, & #6 is now moved to ref. #6, #7, #10, #11, #12, #17, #18, #19, #20, #21, #22, #23, & #24 respectively. Ref. #14 & #16 is newly inserted; 

In Study design and data source (line 105): Ref. #23 is now moved to the ref. #25

In Severity of COVID-19 infection (line 121): Ref. #25 is now moved to the ref. #26

In Statistical analysis (line142): Ref. #26 is now moved to the ref. #27, Ref. #28 is newly inserted; 

In Discussion (line 245): Ref. #28, #29, #27, #31, #32, #33, #34, #35, & #36 is now moved to ref. #29, #30, #31, #32, #33, #34, #35, #36, & #37 respectively.

Disease severity is clearly defined based on the “Living Guidance for clinical management of COVID-19 2021 by WHO (line 121 - 138)”. Now, the disease severity and patient’s characteristics are clearly stated and properly acknowledged in the Result section. Below are the changes:

In the discussion section, I compared and contrasted with the current available evidences. I made the following changes: “In 2020, a research conducted by the Center for Disease Control (CDC) COVID-19 Response Team in USA showed that smokers had higher risk of having critical condition than never-smoker” by “For assessing the impact of smoking on COVID-19 severity, a meta-analysis found that patients who were current or former smoker had an increased risk of severe or critical COVID-19 infection”. I have also updated the references in the main text (line 260 - 264). I have also added the paragraph” “We found that CVD are independently associated with severity of COVID-19 infection, which is found to be insignificant when adjusting with its excoriates. Similar findings were observed in, Kunihiro et al. study that CVD were independently associated with severity of COVID-19” (line 286 - 289). 

Laboratory protocols is not applicable because the samples are collected in the SARI ITC icddr,b and tested at RT PCR laboratory set-up by World Health Organization in Cox’s Bazar as COVID-19 emergency response and is permitted to use for research purpose.

---

## [Decision Letter · Decision Letter 1]

4 Aug 2023

PONE-D-22-33469R1Smoking and Pre-existing Co-morbidities as Risk Factors for Developing Severity of COVID-19 Infection: Evidence from a Field Hospital in a Rural Area of BangladeshPLOS ONE

Dear Dr. Islam,

Thank you for submitting your manuscript to PLOS ONE. After careful consideration, we feel that it has merit but does not fully meet PLOS ONE’s publication criteria as it currently stands. Therefore, we invite you to submit a revised version of the manuscript that addresses the points raised during the review process.

We look forward to receiving your revised manuscript.

Kind regards,

Mosharop Hossian

Academic Editor

PLOS ONE

Journal Requirements:

Reviewers' comments:

Reviewer's Responses to Questions

**Comments to the Author**

1. If the authors have adequately addressed your comments raised in a previous round of review and you feel that this manuscript is now acceptable for publication, you may indicate that here to bypass the “Comments to the Author” section, enter your conflict of interest statement in the “Confidential to Editor” section, and submit your "Accept" recommendation.

Reviewer #1: (No Response)

Reviewer #2: (No Response)

Reviewer #3: (No Response)

Reviewer #4: (No Response)

Reviewer #5: (No Response)

2. Is the manuscript technically sound, and do the data support the conclusions?

Reviewer #1: Partly

Reviewer #2: Yes

Reviewer #3: Partly

Reviewer #4: Partly

Reviewer #5: Yes

3. Has the statistical analysis been performed appropriately and rigorously? 

Reviewer #1: No

Reviewer #2: Yes

Reviewer #3: I Don't Know

Reviewer #4: Yes

Reviewer #5: Yes

4. Have the authors made all data underlying the findings in their manuscript fully available?

Reviewer #1: Yes

Reviewer #2: Yes

Reviewer #3: No

Reviewer #4: Yes

Reviewer #5: Yes

5. Is the manuscript presented in an intelligible fashion and written in standard English?

Reviewer #1: No

Reviewer #2: Yes

Reviewer #3: No

Reviewer #4: Yes

Reviewer #5: No

6. Review Comments to the Author

Reviewer #1: The Manuscript investigates an interesting topic however, the authors did not provide highly considerable finding as follows:

1- The authors did not get any permission from the patients to use their data. Even if the data are de-identified in the hospital archives, patients shall be asked to get their permission as a part of the ethical rights of patients.

2- The authors did not explain the reason behind applying two different statistical tests for association.

3- The age variable is not investigated properly. The association of age groups with respect to COVID severity shall be used according to the WHO classification of age groups by infection (not below 60 and above 60 years old).

4- The discussion section needs an additional improvement to link the finding of the paper with the published data. The authors compared their finding only with a limited number of published studies. Thus, the contradiction of association (mentioned in the introduction) is not addressed again in the discussion to fill this research gap.

Reviewer #2: Thank you very much for the great work on this manuscript.

Just a few comments:

In line 74: “However, patients with ….” Do you mean “However, in other studies patients with ..”? if not please rephrase.

In line 78: it states there are only a couple of studies in Bangladesh as of now

The mentioned one in the manuscript is referenced as [6]: Mohsin FM, Tonmon TT, Nahrin R, Tithy SA, Ame FA, Ara I, et al. Association Between 352 Smoking and COVID-19 Severity: Evidence from Bangladesh. J Multidiscip Healthc 353 [Internet]. 2021 Jul [cited 2021 Aug 26];14:1923. Available from: /pmc/articles/PMC8315768/

What is the other study?

Line 85 and 86: can you rephrase this part? the purpose of the hospital was to manage COVID-19 outbreak from when to when?

Line 90: “…during the study period.” Only during that time?

Line 92: “…supplied as well” and line 94: “…deployed to run ….. ensured ...” by who ? or rephrase please.

172: Severity “of”

334: Conclusion

can you please add specific recommendation for smoking and COVID-19

Thank you for the great work again.

Reviewer #3: The manuscript requires improvement in its English language usage. While the content is satisfactory, the writing needs to be enhanced. The author should focus on refining the methods, results, and conclusion sections.

Reviewer #4: General comments:

The authors present a paper on hospital-level data of COVID-19 patients, their disease severity and the association of disease severity and selected risk factors. The study is carefully thought out and statistically sound. There are a few points that the authors might consider to strengthen the overall contribution.

The title of this paper suggests that the primary association that is under investigation is between COVID-19 disease severity and smoking. However, there is little background on tobacco use behaviors in the country and the level of detailed data collected and analyzed is very superficial.

A stronger rationale for the co-morbidities chosen would help frame the aims of the paper.

Careful proofreading of the flow of English would help with clarity in some parts (eg – Introduction) of the paper. Also, please check spellings of words.

Introduction:

General comments:

The authors present a succinct literature review stating the existing research on COVID-19 risk and associated risk factors. It may help to present some background data on the impact of COVID-19 in the country or region. Also, what is the burden of disease and associated risk factors for CVD and diabetes? It would help to have more on this to frame the issue.

Regarding the use of tobacco, what is the most commonly used form of tobacco? If you have data on frequency or amount smoked per day that would be interesting. Did you account for chewing tobacco use?

Please add citation for statement on line59:

“Smokers are 34% more likely to contract the common flu than non-smokers.”

Methods:

Did you have a pre-specified analysis plan? if so it would be good to state this.

It is not fully clear from your description of your data, or statistical analysis what “factors” you are assessing other than smoking. What are the other co-morbidities and how were they defined (lab values, cut-offs, self-report)?

Results:

The results of this study are not surprising. Did you consider any interaction effects (smoking*CVD or smoking*COPD or Diabetes*CVD)?

Discussion and Conclusion

The data was collected over several waves, and so I wondered if there was any discussion on how vaccination uptake may/may not have moderated the severity of disease? If this was not collected or considered in the analysis, then it should be discussed as a limitation.

I think that the discussion could be expanded a bit to draw a connection to the policy and practice implications stated in the conclusion.

Reviewer #5: Reviewer Comments for "Smoking and Pre-existing Co-morbidities as Risk Factors for Developing Severity of COVID-19 Infection: Evidence from a Field Hospital in a Rural Area of Bangladesh"

Summary

This study is a retrospective analysis of COVID-19 hospital admissions to a rural hospital in Bangladesh. It is unique in its description of its target population in a rural, low-resource setting. It is commendable to find detailed hospital records that could test the association between co-morbidities like COPD, Hypertension, Asthma and CVD with the propensity to develop moderate, severe and critical COVID-19 infection. It also tested the association between demographic characteristics like age, gender and risk habits like smoking with COVID-19 severity. The authors successfully tested the relationship using bivariate analysis (Chi-square analysis and Fisher's exact test) and multivariate analysis (Multinomial logistic regression). The results were presented in clear and concise tables. This study is precise in its aims and has successfully met them. The following comments could improve the study's reporting and enhance the paper's quality.

Abstract

1. Line # 23: "Based on inclusion criteria;" seems redundant. You could say that "788

admitted patients were included in the analysis."

2. Line # 34: A measure of association to the result would increase its credibility –"It was

evident that the patients with at least one of the selected co-morbidities (e.g.,

hypertension, diabetes, COPD, CVD, asthma) had suffered severe illness of COVID-19

compared to the patients who had no co-morbidity."

Background

The background clearly describes COVID-19 infection and how co-morbidities influence it.

Materials and Methods

1. Line 85-96: The study setting gives a detailed description of the field hospital in Teknaf. A brief description of the catchment area/patient population would enhance this description. One or two lines on the patient population - regarding their socio-economic status, dominant occupation, etc.- will help set the study context.

2. Line 153-154: To improve the multinomial logistic regression model reporting, you could mention the independent variables (predictors) used in the analysis. To be more specific, listing the control variables used in the model that examines the "adjusted relationship" between the severity of COVID-19 infection and smoking could help understand the regression model.

3. Line 141: If a categorical variable like the Severity of Covid-19 infection is used as the dependent variable, it would help to mention the reference category and why it was used as a reference category. The justification could be linked to the implications of the result if possible.

4. Line 154: You could include measures of model fit to assess how well the model fits the data. Common goodness-of-fit statistics for multinomial logistic regression include the likelihood ratio chi-square test, the McFadden's pseudo R-squared, or the AIC (Akaike Information Criterion). It is mentioned in the results section but can be mentioned in the methodology section.

5. Could you justify why current and past smokers were combined as one variable? Was it because of uneven distribution across the two categories? Or was it because of a lack of differentiation between the two categories in the database?

6. Were there any missing data or incomplete records?

Results

1. Line 170: The title of the table could be more elaborate. You could detail what kind of characteristics-demographic/ clinical you are showing. In addition, you could specify what kind of patients you are describing-COVID-19 patients admitted to the field hospital. A detailed title will help quick readers skim through your results.

2. Line 179: Table 2 shows the distribution of variables in addition to association. Could you highlight that in your title?

3. Line 205: You could list what the other variables/confounders were added to your final model.

4. Line 207-271: Please rework reporting the RRR interpretation. For example, in Line 208: It would make sense if you could say that hypertensive patients were 4.75 times more likely to develop critical severity of COVID-10 than non-hypertensive patients. The outcome and the comparator group must be reported together. The different categories must not be compared with each other unless the analysis puts one of them as the reference category.

5. Line 217: Please specify the comparator group(those not suffering from COPD).

6. Line 218: Please specify the comparator group similar to how you have described in line 224.

7. Line 223: RRR shows increased odds of developing moderate/severe/critical COVID-19 infection from the reference mild COVID-19 infection. You could report … "more risks higher odds of reporting critical, severe, and moderate COVID-19 infected patients, infections than mild COVID-19 infection, when compared to adults …Reference category has to be specified when interpreting the relative risk ratio.

Discussion

1. Line 303: Can you rule out misclassification errors that could occur if the patient did not know if they had CVD, COPD or hypertension prior to COVID-19 infection? Were they tested for all these co-morbidities in the hospital? Were these co-morbidities diagnosed prior to the admission (hospital records of the only facility in the area) or during admission (first-time patients, refugees with no prior records) etc.

2. If you could describe your study population, the results could be generalised to this group. This could be a strength of your study, as it describes the particular group.

3. Line 311: As you mentioned that a significant proportion of the population is unaware of their co-morbidities, you could suggest mass awareness campaigns should be coupled with diagnostic infrastructure the target population could access.

Acknowledgement

I am glad that the authors have acknowledged the support staff and managers who make hospital records a rich resource for researchers. To be able to extract data without missing reports or incomplete reports is a feat.

7. PLOS authors have the option to publish the peer review history of their article (what does this mean?). If published, this will include your full peer review and any attached files.

Reviewer #1: No

Reviewer #2: No

Reviewer #3: No

Reviewer #4: No

Reviewer #5: **Yes: **Preetha Menon

---

## [Author Response · Author response to Decision Letter 1]

11 Aug 2023

Dear Mosharop Hossian,

We would like to express our sincere gratitude to you and the reviewers for taking the time to assess our manuscript titled "Smoking and Pre-existing Co-morbidities as Risk Factors for Developing Severity of COVID-19 Infection: Evidence from a Field Hospital in a Rural Area of Bangladesh." We value the constructive feedback provided by the reviewers, and we have carefully addressed each point raised in the reviews. 

Please find our detailed responses below, along with the corresponding changes made in the revised manuscript.

Reviewer #1: The Manuscript investigates an interesting topic however; the authors did not provide highly considerable findings as follows:

Comment 1- The authors did not get any permission from the patients to use their data. Even if the data are de-identified in the hospital archives, patients shall be asked to get their permission as a part of the ethical rights of patients.

Response: Thank you for your feedback. In the ethical clearance section, we mention that the participant provided verbal consent. In order to make it more understandable, we have rephrased it.

Comment: 2- The authors did not explain the reason behind applying two different statistical tests for association.

Response: We appreciate your comment regarding the statistical tests. The decision to apply two different statistical tests, Chi-square test and Fisher's exact test, for association was based on established statistical principles. While the Chi-square test is commonly used for identifying associations between categorical variables, it may yield less reliable results when the Expected cell frequency is less than 5. In such cases, Fisher's exact test provides a more robust measure. In this study, we found that the Expected cell frequency was less than 5 for certain variables, including Age, Diabetes, COPD, CVD, and Asthma. Hence, we opted for Fisher's exact test to ensure a more accurate and reliable analysis for these specific variables.

Comment: 3- The age variable is not investigated properly. The association of age groups with respect to COVID severity shall be used according to the WHO classification of age groups by infection (not below 60 and above 60 years old).

Response: Thank you for your valuable feedback. We acknowledge the importance of using age groups based on the WHO classification for COVID-19 severity analysis. Regarding the age variable, we chose to categorize patients as "adults" (18-60 years) and "elderly" (>60 years) due to the observed significant association between these age groups and the severity of COVID-19 infection in the pilot analysis.

Comment: 4- The discussion section needs an additional improvement to link the finding of the paper with the published data. The authors compared their findings only with a limited number of published studies. Thus, the contradiction of association (mentioned in the introduction) is not addressed again in the discussion to fill this research gap.

Response: Thank you for your suggestions. Based on your suggestions, we have now enhanced the discussion section by adding a few references addressing the contradiction of association.

Reviewer #2: Thank you very much for the great work on this manuscript. Just a few comments:

Comment: In line 74: “However, patients with ….” Do you mean “However, in other studies patients with.”? if not please rephrase.

Response: We appreciate the comment. We have rephrased the sentence in line 74 to clarify the statement with a proper citation which now can be found on line number 98 to 100: " However, a study conducted on symptomatic patients found that COPD and asthma alone did not lead to worsening infections, hospitalizations, or deaths associated with COVID-19 [30]

Comment: In line 78: it states there are only a couple of studies in Bangladesh as of now The mentioned one in the manuscript is referenced as [6]: Mohsin FM, Tonmon TT, Nahrin R, Tithy SA, Ame FA, Ara I, et al. Association Between 352 Smoking and COVID-19 Severity: Evidence from Bangladesh. J Multidiscip Healthc 353 [Internet]. 2021 Jul [cited 2021 Aug 26];14:1923. Available from: /pmc/articles/PMC8315768/ What is the other study?

Response: We appreciate your keen observation and valuable input. Since we have only found one study as of now, we have rephrased the statement in line 78, which can be found in lines 102 to 104, as follows: " Despite several studies in many countries, to date, only one study has been found in Bangladesh on the associated risk factors for severe COVID-19 infection, such as smoking [6] and other co-morbidities."

Comment: Line 85 and 86: can you rephrase this part? the purpose of the hospital was to manage COVID-19 outbreak from when to when?

Response: We appreciate the reviewer's comment and helpful suggestion for rephrasing. The revised version of the sentence, now found on line number 92, reads: "As part of the COVID-19 outbreak management, operations began on August 27, 2020, and are still in progress"

Comment: Line 90: “…during the study period.” Only during that time? Line 92: “…supplied as well (by the donor)” and line 94: “…deployed to run ….. ensured ...” by who ? or rephrase please.

Response: Thank you so much, I really appreciate it. The free of charge is not only during the study period but still in progress. The materials were supplied by the Donor and deployed by the ICDDR, B. The following points are now clearly stated to make it clearer, as shown in line numbers 116-121.

Comment: 172: Severity “of”, 334: Conclusion can you please add specific recommendations for smoking and COVID-19 Thank you for the great work again.

Response: Thank you for pointing this out. The missing preposition "of" has now been added to new line 191 and the formatting mark has been removed from your mentioned line number, which is now line 370. We have also added the following specific recommendations which can be found in lines number 378 and 384, read as: “It is recommended that public health interventions should prioritize smoking cessation programs and awareness campaigns to reduce smoking prevalence among the population, particularly those with comorbidities. Furthermore, efforts should be made to enhance the early detection and management of hypertension, diabetes, COPD, and asthma in the community to mitigate the severity of COVID-19 infections and improve overall health outcomes.”

Reviewer #3, Comment: The manuscript requires improvement in its English language usage. While the content is satisfactory, the writing needs to be enhanced. The author should focus on refining the methods, results, and conclusion sections.

Response: Thank you for acknowledging the satisfactory content of our manuscript. We have taken your comments seriously and have focused on refining the English language usage in the methods, results, and conclusion sections to ensure better readability and clarity. We believe that these improvements will enhance the overall presentation of our findings and make them more accessible to the readers. Your valuable input has been instrumental in improving the quality of our manuscript, and we are grateful for your guidance.

Reviewer #4: General comments: The authors present a paper on hospital-level data of COVID-19 patients, their disease severity and the association of disease severity and selected risk factors. The study is carefully thought out and statistically sound. There are a few points that the authors might consider to strengthen the overall contribution.

Comment: The title of this paper suggests that the primary association that is under investigation is between COVID-19 disease severity and smoking. However, there is a little background on tobacco use behaviors in the country, and the level of detailed data collected and analyzed is very superficial.

Response: Thank you for your feedback. We have enhanced our background by adding the following details “Bangladesh is significant, with nearly 35% of adults smoking or smokeless tobacco currently. Since 2004, this lower middle-income country has witnessed a dramatic increase in the health and economic burden linked to tobacco use, as evidenced by primary data from a nationally representative survey of patients with tobacco-related illnesses. Tobacco smoking significantly contributes to noncommunicable diseases (NCDs), including cardiovascular and respiratory conditions, underscoring the imperative to address tobacco control measures to mitigate its impact on both public health and economic growth.”

Comment: A stronger rationale for the co-morbidities chosen would help frame the aims of the paper.

Response: Rational about some co-morbidities have already been addressed, and the rationale about diabetes as a co-morbidity has been newly added and can be read as” Diabetic patients with COVID-19 have a more severe condition and poorer clinical outcomes, including a higher mortality rate and a higher ICU admission rate, highlighting the necessity for better management strategies in order to improve their prognosis.”

Comment: Careful proofreading of the flow of English would help with clarity in some parts (eg – Introduction) of the paper. Also, please check spelling of words.

Response: Thank you for your feedback. We have now carefully proofread the manuscript and corrected any misspelled words found.

Comment: Introduction: General comments: The authors present a succinct literature review stating the existing research on COVID-19 risk and associated risk factors. It may help to present some background data on the impact of COVID-19 in the country or region. Also, what is the burden of disease and associated risk factors for CVD and diabetes? It would help to have more on this to frame the issue.

Response: Thank you for your feedback. We have background data on the impact of COVID-19 in the country (Bangladesh), and this can be found in line # as “. In Bangladesh, the first 3 COVID-19 cases were detected on March 2020[4]and more than 2.0 million COVID-19 cases were identified, including about 29,300 deaths till August 10, 2022”. Although the burden of the co-morbidities like CVD and Diabetes were not our primary focus of the study, however, we have added a background related to burden of diabetes in the country in the introduction section.

Comment: Regarding the use of tobacco, what is the most commonly used form of tobacco? If you have data on the frequency or amount smoked per day that would be interesting. Did you account for chewing tobacco use?

Response: Thank you for your question. All the patient included in the study verbally acknowledged that they were smoking cigarette. The study did not specifically collect data on the frequency or amount smoked per day. Additionally, chewing tobacco use was not accounted for in the analysis, as the study focused primarily on the association between smoking and COVID-19 severity. However, we have included a clarification in the manuscript to explicitly mention that the study focused on cigarette smoking.

Comment: Please add citation for statement on line59: “Smokers are 34% more likely to contract the common flu than non-smokers.”

Response: Thank you for pointing this out. We have now added the citation and can be found on the new line number # 60

Comment: Methods: Did you have a pre-specified analysis plan? if so it would be good to state this. It is not fully clear from your description of your data, or statistical analysis what “factors” you are assessing other than smoking. What are the other co-morbidities and how were they defined (lab values, cut-offs, self-report)?

Response: Thank you for your question. The following were considered in the Analysis plan: “ COPD, hypertension, cardiovascular disease, and diabetes were diagnosed based on patient history, clinical features, and radiological and laboratory findings. The severity of COVID-19 infection was determined based on WHO standards. Descriptive statistics were used to determine the frequency and percentage of background, characteristics, severity levels of COVID-19 infections, pre-existing comorbidities, and smoking history. To assess the association between main variables of interest (e.g. history of smoking, underlying co-morbidities) and severity of COVID-19 infection, Pearson's Chi-square test and Fisher's exact test are used. Multinomial logistic regression models were used to identify potential risk factors and determine the effect size of these factors on the progression of COVID-19 infection severity in the patients. To measure statistical significance, a P-value of 0.05 was considered.”

Comment: Results: The results of this study are not surprising. Did you consider any interaction effects (smoking*CVD or smoking*COPD or Diabetes*CVD)?

Response: Thank you for your question. The study primarily focused on assessing the individual effects of various risk factors on the severity of COVID-19 infection. While the potential for interaction effects, such as smoking interacting with CVD or COPD, and diabetes interacting with CVD, could provide valuable insights, these specific interaction effects were not investigated in this study. The focus was primarily on understanding the independent contributions of each factor to the severity of COVID-19 infection.

Discussion and Conclusion

Comment: The data was collected over several waves, and so I wondered if there was any discussion on how vaccination uptake may/may not have moderated the severity of disease? If this was not collected or considered in the analysis, then it should be discussed as a limitation. I think that the discussion could be expanded a bit to draw a connection to the policy and practice implications stated in the conclusion.

Response: Thank you for your insightful feedback. As a result of your feedback, we have mentioned our limitation as 'The vaccination uptake and its potential role in moderating the severity of COVID-19 infection were not considered in the analysis'. Also, the clinical team did not perform spirometry to diagnose COPD.”Furthermore, to address your point, we have expanded our conclusion and it reads as follows: Also, it is recommended that public health interventions should prioritize smoking cessation programs and awareness campaigns to reduce smoking prevalence among the population, particularly those with comorbidities. Among the vulnerable group of people, the implementation of mass awareness campaigns along with diagnostic infrastructure concerning pre-existing co-morbidities may result in a reduction of severity and mortality rate.”

Reviewer #5: Reviewer Comments for "Smoking and Pre-existing Co-morbidities as Risk Factors for Developing Severity of

COVID-19 Infection: Evidence from a Field Hospital in a Rural Area of Bangladesh"

Summary

This study is a retrospective analysis of COVID-19 hospital admissions to a rural hospital in Bangladesh. It is unique in its description of its target population in a rural, low-resource setting. It is commendable to find detailed hospital records that could test the association between co-morbidities like COPD, Hypertension, Asthma and CVD with the propensity to develop moderate, severe and critical COVID-19 infection. It also tested the association between demographic characteristics like age, gender and risk habits like smoking with COVID-19 severity. The authors successfully tested the relationship using bivariate analysis (Chi-square analysis and Fisher's exact test) and multivariate analysis (Multinomial logistic regression).

The results were presented in clear and concise tables. This study is precise in its aims and has successfully met them. The following comments could improve the study's reporting and enhance the paper's quality.

Abstract

Comment: 1. Line # 23: "Based on inclusion criteria;" seems redundant. You could say that "788 admitted patients were included in the analysis."

Response: We appreciate your suggestion. Now, we have made this change and can be found in line number #137

Comment: 2. Line # 34: A measure of association to the result would increase its credibility –"It was evident that the patients with at least one of the selected co-morbidities (e.g., hypertension, diabetes, COPD, CVD, asthma) had suffered severe illness of COVID-19 compared to the patients who had no co-morbidity."

Response: Thank you so much. We have made the necessary adjustment to the sentence you mentioned: "It was evident that patients with at least one of the selected co-morbidities (such as hypertension, diabetes, COPD, CVD, asthma) exhibited a significantly higher likelihood of experiencing severe illness due to COVID-19 compared to patients without any co-morbidity." This revision emphasizes the statistically significant association between the presence of co-morbidities and the severity of COVID-19 infection, thus strengthening the credibility of our results. You can find this on new line numbers 37-41

Background

The background clearly describes COVID-19 infection and how co-morbidities influence it.

Materials and Methods

Comment: 1. Line 85-96: The study setting gives a detailed description of the field hospital in Teknaf. A brief description of the catchment area/patient population would enhance this description. One or two lines on the patient population - regarding their socio-economic status, dominant occupation, etc.- will help set the study context.

Response: We greatly appreciate your feedback and suggestion to provide more context regarding the catchment area and patient population. While we acknowledge the importance of socio-economic status and its potential impact on the study findings, we regret to inform you that we did not include participants' socio-economic status in our analysis due to the emergency of Covid-19 pandemic at hospital setup. As you correctly pointed out, our analysis primarily focused on factors such as age, sex, smoking history, and underlying co-morbidities in relation to the severity of COVID-19 infection which are considered highly relevant to the COVID-19 severity.

Comment: 2. Line 153-154: To improve the multinomial logistic regression model reporting, you could mention the independent variables (predictors) used in the analysis. To be more specific, listing the control variables used in the model that examines the "adjusted relationship" between the severity of COVID-19 infection and smoking could help understand the regression model.

Response: Thank you for your valuable feedback. We have addressed your point and rephrased the sentence by adding the control variables and can be found in new line numbers #184-#188 as” The following equation (2) was estimated to examine the relationship between the severity of COVID-19 infection and the history of smoking with control variables, such as age, sex, hypertension, diabetes, COPD, CVD, and asthma.”

Comment: 3. Line 141: If a categorical variable like the Severity of Covid-19 infection is used as the dependent variable, it would help to mention the reference category and why it was used as a reference category. The justification could be linked to the implications of the result if possible.

Response: Multinomial logistic regression models [27] were used to identify the potential risk factors and measure the effect size of these factors on the progression of severity of COVID-19 infection in the patients where "mild case" was used as a reference category’ because severity increases from mild to critical when the factors of interest are present. We have now rephrased and updated the manuscript.

Comment: 4. Line 154: You could include measures of model fit to assess how well the model fits the data. Common goodness-of-fit statistics for multinomial logistic regression include the likelihood ratio chi-square test, the McFadden's pseudo R-squared, or the AIC (Akaike Information Criterion). It is mentioned in the results section but can be mentioned in the methodology section.

Response: The following equation (2) was estimated to examine the adjusted relationship between the severity of COVID-19 infection and the history of smoking with control variables. We mentioned the likelihood ratio chi-square test and McFadden's pseudo-R-squared values in the results section for assessing the fit of the model to the data.

Comment: 5. Could you justify why current and past smokers were combined as one variable? Was it because of uneven distribution across the two categories? Or was it because of a lack of differentiation between the two categories in the database?

Response: Thank you for bringing this to our attention. Our primary focus was to assess the impact of smoking on the severity of COVID-19, rather than evaluating its duration. While there was a distinction between the two categories in our database, the differentiation was based on verbal acknowledgment from the patients. Collecting concrete evidence regarding a patient's smoking status, particularly in the context of the COVID-19 situation, is challenging and often relies on verbal consent due to the limitations posed by the circumstances.

Comment: 6. Were there any missing data or incomplete records?

Response: Thank you. No missing data or incomplete records were included in the study. We extracted data that aligned with our predefined inclusion criteria, ensuring a comprehensive and complete dataset for our analysis.

Results

Comment: 1. Line 170: The title of the table could be more elaborate. You could detail what kind of characteristics-demographic/ clinical you are showing. In addition, you could specify what kind of patients you are describing-COVID-19 patients admitted to the field hospital. A detailed title will help quick readers skim through your results.

Response: Thank you for your valuable feedback. We appreciate your suggestion for a more informative and comprehensive title for the table. We have updated the title as per your recommendation to enhance clarity and aid quick readers in comprehending the content. The revised title now reads “Table 1: Distribution of Patient Characteristics and Co-morbidities in COVID-19 Cases”. We hope this revised title effectively communicates the scope of the table's contents and provides a clear overview for readers.

Comment: 2. Line 179: Table 2 shows the distribution of variables in addition to association. Could you highlight that in your title?

Response: Thank you for your valuable input. We have revised the title of the table to accurately reflect its contents. The new title is: “Table 2: Distribution and Association of Patient Characteristics with Severity of COVID-19 Infection” which highlights both the distribution of patient characteristics and their association with the severity of COVID-19 infection, as indicated in the table.

Comment: 3. Line 205: You could list what the other variables/confounders were added to your final model.

Response: Thank you for pointing out this. We have now added the other confounders included in the final model and rephrased them as follows” As potential risk factors, the following variables were incorporated into the final model (adjusted model): smoking history, hypertension, diabetes, COPD, CVD, asthma, gender, and age (>60 years old - elderly). In the adjusted model, while smoking history was recognized as a risk factor contributing to the severity of COVID-19 infection, it was not associated with moderate and critical levels of severity.” You can find this on the new line number # 246-251

Comment: 4. Line 207-271: Please rework reporting the RRR interpretation. For example, in Line 208: It would make sense if you could say that. The outcome and the comparator group must be reported together. The different categories must not be compared with each other unless the analysis puts one of them as the reference category. 

5. Line 217: Please specify the comparator group(those not suffering from COPD).

6. Line 218: Please specify the comparator group similar to how you have described in line 224.

7. Line 223: RRR shows increased odds of developing moderate/severe/critical COVID-19 infection from the reference mild COVID-19 infection. You could report … "more risks higher odds of reporting critical, severe, and moderate COVID-19 infected patients, infections than mild COVID-19 infection, when compared to adults …Reference category has to be specified when interpreting the relative risk ratio.

Responses (4, 5, 6, 7): Thank you very much for your valuable suggestion. The sentences have now been rephrased based on your feedback and can be found on new lines 250- 270

Discussion

Comment: 1. Line 303: Can you rule out misclassification errors that could occur if the patient did not know if they had CVD, COPD or hypertension prior to COVID-19 infection? Were they tested for all these co-morbidities in the hospital? Were these comorbidities diagnosed prior to the admission (hospital records of the only facility in the area) or during admission (first time patients, refugees with no prior records) etc.

Response: Thank you for your feedback. While the patients in our study were tested clinically and laboratory confirmation for these co-morbidities during their hospital stay, there is a possibility that some patients may not have been aware of their pre-existing conditions. This particularly applies to first-time patients, like the Forcibly Displaced Myanmar Nationals (FDMN) refugees, who might lack prior medical records. These points have also been addressed and rephrased in the manuscript and can be found on lines 138-140 in the methodology section.

Comment: 2. If you could describe your study population, the results could be generalized to this group. This could be a strength of your study, as it describes the particular group.

Response: We appreciate the idea. This study population can be found in the study setting section, which states that "The hospital is a not-for-profit organization serving locals, and Forcibly Displaced Myanmar Nationals (FDMN) in Teknaf."

Comment: 3. Line 311: As you mentioned that a significant proportion of the population is unaware of their co-morbidities, you could suggest mass awareness campaigns should be coupled with diagnostic infrastructure the target population could access.

Response: Thank you for sharing your thoughts. Our conclusion section now discusses the need to link mass awareness campaigns with diagnostic infrastructure and can be found on line 378-380.

Acknowledgment

Comment: I am glad that the authors have acknowledged the support staff and managers who make hospital records a rich resource for researchers. To be able to extract data without missing reports or incomplete reports is a feat.

Response: Yes, acknowledging the staff behind the data is an honor and we appreciate your valuable comments.

In addition to addressing the specific points raised by the reviewers, we have carefully proofread and improved the manuscript language to correct any grammatical errors and ensure clarity. We believe that these revisions significantly enhance the quality and impact of the manuscript.

Once again, we are grateful for the opportunity to revise and resubmit our work. We sincerely hope that the changes made in the manuscript adequately address the reviewers' concerns and meet the standards of the [Journal Name]. We eagerly await further guidance and feedback from you and the reviewers.

Sincerely,

Md. Rashadul Islam

---

## [Decision Letter · Decision Letter 2]

3 Nov 2023

PONE-D-22-33469R2Smoking and Pre-existing Co-morbidities as Risk Factors for Developing Severity of COVID-19 Infection: Evidence from a Field Hospital in a Rural Area of BangladeshPLOS ONE

Dear Dr. Islam,

Thank you for submitting your manuscript to PLOS ONE. After careful consideration, we feel that it has merit but does not fully meet PLOS ONE’s publication criteria as it currently stands. Therefore, we invite you to submit a revised version of the manuscript that addresses the points raised during the review process.

The manuscript has been evaluated by three of  the reviewers who provided feedback on the previous version of your paper, and their comments are available below. The reviewers are satisfied with the revisions you made in response to their comments, but they have a few requests for minor amendments. Could you please revise the manuscript to carefully address the concerns raised?

We look forward to receiving your revised manuscript.

Kind regards,

Steve Zimmerman, PhD

Senior Editor, PLOS ONE

Journal Requirements:

Reviewers' comments:

Reviewer's Responses to Questions

**Comments to the Author**

1. If the authors have adequately addressed your comments raised in a previous round of review and you feel that this manuscript is now acceptable for publication, you may indicate that here to bypass the “Comments to the Author” section, enter your conflict of interest statement in the “Confidential to Editor” section, and submit your "Accept" recommendation.

Reviewer #2: All comments have been addressed

Reviewer #4: (No Response)

Reviewer #5: All comments have been addressed

2. Is the manuscript technically sound, and do the data support the conclusions?

Reviewer #2: Yes

Reviewer #4: Yes

Reviewer #5: Yes

3. Has the statistical analysis been performed appropriately and rigorously? 

Reviewer #2: I Don't Know

Reviewer #4: Yes

Reviewer #5: Yes

4. Have the authors made all data underlying the findings in their manuscript fully available?

Reviewer #2: Yes

Reviewer #4: Yes

Reviewer #5: Yes

5. Is the manuscript presented in an intelligible fashion and written in standard English?

Reviewer #2: Yes

Reviewer #4: Yes

Reviewer #5: Yes

6. Review Comments to the Author

Reviewer #2: Thank you for addressing the previous comments and the great work on the manuscript language.

Minor comments:

In line 97, the word "hospitalization" is duplicated, please delete.

from line 110 to 112: please rephrase this to be more understandable. For example: It used to serve as ... and since August 27, 2020, it began to serve as .. and continue until now.

In line 116: since the hospital, it's services and supplies still continue to serve people for free, please adjust the grammar of the sentence accordingly.

In line 137: can you please clarify if this sentences is being referenced to reference number [32] and what information the reference is supporting? [32]: if not please check your references, as reference [32] is only mentioned in this line.

According to your response to the reviewer, you stated that "We chose to categorize patients as "adults" (18-60 years) and "elderly" (>60 years) due to the observed significant association between these age groups and the severity of COVID-19 infection in the pilot analysis." I would suggest adding this information to the manuscript.

line 380: please rephase.

Reviewer #4: Thank you for your revision. I find the manuscript much improved. I only have a few comments for the authors to address:

Background, line 105-106: "...either cigarettes or other tobacco products..." Only cigarette smoking is addressed in the manuscript and so if other tobacco products were not considered it should be noted as a limitation or the reference to these should be removed.

Methods, line137: remove (why was this reference deleted?)

Results, line 250: word choice "riskier" is awkward

Discussion:

You did not analyze the effects of sex and smoking combined, and it is noted here in your manuscript, but it should be noted as a limitation as there may be an interaction effect on men being more likely to develop severe COVID due to more of them being smokers.

What do you make of the non-significance of the association between smoking status and critical COVID in the adjusted model?

line 301: "consistence" should be "consistent"

Reviewer #5: (No Response)

7. PLOS authors have the option to publish the peer review history of their article (what does this mean?). If published, this will include your full peer review and any attached files.

Reviewer #2: No

Reviewer #4: No

Reviewer #5: No

---

## [Author Response · Author response to Decision Letter 2]

4 Nov 2023

Date: 04.11.2023

Subject: Response to Reviewers - Manuscript Number PONE-D-22-33469R1.

Dear Steve Zimmerman,

We would like to express our sincere gratitude to you and the reviewers for taking the time to assess our manuscript titled "Smoking and Pre-existing Co-morbidities as Risk Factors for Developing Severity of COVID-19 Infection: Evidence from a Field Hospital in a Rural Area of Bangladesh." We value the constructive feedback provided by the reviewers, and we have carefully addressed each point raised in the reviews.

Please find our detailed responses below, along with the corresponding changes made in the revised manuscript.

Reviewer #2: Thank you for addressing the previous comments and the great work on the manuscript language.

Reviewer comment #1: In line 97, the word "hospitalization" is duplicated, please delete.

Author repones: Thank you. Now the duplicate has been removed.

Reviewer comment #2: from line 110 to 112: please rephrase this to be more understandable. For example: It used to serve as ... and since August 27, 2020, it began to serve as .. and continue until now.

Author response: We appreciate your feedback. It has now been rephrased as ” The hospital, initiated as part of COVID-19 outbreak management in Teknaf, began operations on August 27, 2020, and it is still actively providing services to the local population and Forcibly Displaced Myanmar Nationals (FDMN) as a not-for-profit organization” 

Reviewer comment #3: In line 116: since the hospital, it's services and supplies still continue to serve people for free, please adjust the grammar of the sentence accordingly.

Author response: We've corrected the grammar, and it now reads “Besides; 24/7 Emergency, Out-Patient Department, radiology, laboratory services, and referral by ambulance to higher level facilities in Cox’s Bazar district remain available free of charge”

Reviewer comment #4: In line 137: can you please clarify if this sentences is being referenced to reference number [32] and what information the reference is supporting? [32]: if not, please check your references, as reference [32] is only mentioned in this line. 

According to your response to the reviewer, you stated that "We chose to categorize patients as "adults" (18-60 years) and "elderly" (>60 years) due to the observed significant association between these age groups and the severity of COVID-19 infection in the pilot analysis." I would suggest adding this information to the manuscript.

Author response: Thank you for pointing this out. There was an unnecessary text, and it has now been removed. Your suggestions have also been incorporated into the manuscript

Reviewer comment #5: line 380: please rephase.

Author response: We have revised the line (new line #386), and it now reads: “Also, it is recommended that public health initiatives prioritize extensive smoking cessation programs and awareness campaigns aimed at reducing smoking rates, particularly among individuals with pre-existing health conditions. Additionally, enhancing diagnostic infrastructure for those with comorbidities can aid in the timely identification of their health issues.” 

Reviewer #4: Thank you for your revision. I find the manuscript much improved. I only have a few comments for the authors to address:

Reviewer comment #1: Background, line 105-106: "...either cigarettes or other tobacco products..." Only cigarette smoking is addressed in the manuscript and so if other tobacco products were not considered it should be noted as a limitation or the reference to these should be removed.

Author response: We appreciate your attention to this matter. We have now clarified in the methodology section (line number #164) that our study considers both cigarette smoking and other tobacco product use as part of the broader category of smoking. This approach allows us to encompass a wide range of tobacco consumption habits.

Reviewer comment #2: Methods, line137: remove (why was this reference deleted?)

Author response: Thank you. Now it has been removed

Reviewer comment #3: Results, line 250: word choice "riskier" is awkward

Author response: We have changed the word "riskier" to "more prone".

Reviewer comment #4: Discussion: You did not analyze the effects of sex and smoking combined, and it is noted here in your manuscript, but it should be noted as a limitation as there may be an interaction effect on men being more likely to develop severe COVID due to more of them being smokers.

Author response: Thank you for your feedback. The prevalence of smoking, which may influence the severity of COVID-19, was not examined separately for males and females in our study. We acknowledged this limitation in the "Limitations" section (Line number #375), highlighting that we did not analyze the potential interaction effects between sex and smoking that could influence the development of severe COVID-19.

Reviewer comment #5: What do you make of the non-significance of the association between smoking status and critical COVID in the adjusted model?

Author Response: Thank you for your question. The non-significance of the "Critical" category in the adjusted model is likely influenced by the smaller sample size of critical cases compared to severe cases. Additionally, the presence of other significant risk factors, such as hypertension, COPD, and age, in the model may have minimized the influence of smoking status when it comes to critical cases. 

Reviewer comment #6: line 301: "consistence" should be "consistent"

Author Response #6: Thank you. We have changed the word.

Sincerely,

Md. Rashadul Islam

---

## [Editor Report · Decision Letter 3]

15 Nov 2023

Smoking and Pre-existing Co-morbidities as Risk Factors for Developing Severity of COVID-19 Infection: Evidence from a Field Hospital in a Rural Area of Bangladesh

PONE-D-22-33469R3

Dear Dr. Islam,

We’re pleased to inform you that your manuscript has been judged scientifically suitable for publication and will be formally accepted for publication once it meets all outstanding technical requirements.

Kind regards,

Steve Zimmerman, PhD

Senior Editor, PLOS ONE
---

## [Editor Report · Acceptance letter]

30 Nov 2023

PONE-D-22-33469R3 

Smoking and Pre-existing Co-morbidities as Risk Factors for Developing Severity of COVID-19 Infection: Evidence from a Field Hospital in a Rural Area of Bangladesh 

Dear Dr. Islam:

I'm pleased to inform you that your manuscript has been deemed suitable for publication in PLOS ONE. Congratulations! Your manuscript is now with our production department. 

Kind regards, 

on behalf of

Dr Steve Zimmerman 

Staff Editor

PLOS ONE